# Effects of Simulated Laughter Therapy Using a Breathing Exercise: A Study on Hospitalized Pulmonary Tuberculosis Patients

**DOI:** 10.3390/ijerph191610191

**Published:** 2022-08-17

**Authors:** Kwang-Sim Jang, Jeong-Eun Oh, Gyeong-Suk Jeon

**Affiliations:** 1Jeon-Nam-Seobu Workers Health Center, Youngam-gun 58454, Korea; 2Mokpo National Hospital, Mokpo 58605, Korea; 3Department of Nursing, Division of Natural Science, Mokpo National University, Muan-gun 58534, Korea

**Keywords:** simulated laughter therapy, tuberculosis, pulmonary function, depression, health-related quality of life

## Abstract

This study evaluated the effects of simulated laughter therapy on physical symptoms, pulmonary function, depression, and health-related quality of life (HRQOL) among pulmonary tuberculosis patients. This quasi-experimental study assigned tuberculosis patients of hospital A to a laughter group (*n* = 26) and those of hospital B to a control group (*n* = 26). The eight-week laughter therapy, held twice a week in a 60-min group session, included laughter, entertainment, music-related chorusing, breathing exercises, and meditation. The values of physical symptoms, pulmonary function, depression, and HRQOL from before and after the therapy were analyzed using the paired *t*-test and the Mann–Whitney U-test. To verify group differences between the experiment and control group, the Wilcoxon signed-rank test and the analysis of covariance (ANCOVA) were employed. Unlike the control group, laughter therapy decreased physical symptoms (t = 7.30, *p* < 0.01) and increased pulmonary function (t = −3.77, *p* < 0.01). Psychological health also improved, including depression (t = 10.46, *p* < 0.01) and HRQOL (t = −9.31, *p* < 0.01) in the experimental group but not in the control group. Group differences of changes in physical symptoms, pulmonary function, depression, and HRQOL were also significant. Simulated laughter therapy can help moderate depression and physical symptoms and enhance pulmonary function among tuberculosis patients.

## 1. Introduction

Tuberculosis (TB) is the leading infectious killer with about one-fourth of the world’s population having latent TB, 9.9 million people suffering from TB, and 1.5 million dying from the disease in 2020 [1]. The Republic of Korea still has the highest tuberculosis incidence rate and the third-highest tuberculosis mortality rate among all member countries of Organization for Economic Co-operation and Development (OECD), even though there has been a rapid decrease since the peak of the Korean War in 1950 [2]. The impact of TB on individuals has far-reaching consequences, not only for their physical health but also on social and psychological well-being [3]. TB patients are severed from their relationships with family, friends, and others, along with economic activities, due to hospitalization, and the stigma of TB may trigger psychological problems, such as stress, anxiety, and depression [4]. Furthermore, patients with TB experience various symptoms, such as dyspnea, cough, sputum, fatigue, loss of appetite, general lethargy, hemoptysis, burning sensation, and side effects of drugs [5]. Mycobacterium tuberculosis induces lung inflammation and destroys lung cells, weakening pulmonary functions [5]. Various physical symptoms, adverse drug reactions (ADRs), and weakened pulmonary functions in patients with TB hinder their activities of daily living, increase depression, and undermine self-esteem, impairing their quality of life (QOL) [4,6]. According to a Korean study, 44–56% of inpatients with TB experience depression [6], which is 1.5 times more than in the general population [7] and patients with cancer and stroke [6]. TB is also one of the major diseases that reduce QOL [8] and patients with TB face sequelae from lung injury even after finishing treatment; they have difficulty restoring the QOL to the level of the general population [4]. In addition, the physical and psychological concerns faced by patients with TB are closely linked to one another [9]. Nevertheless, the current tuberculosis intervention focuses on the mycobacterial cure [1,3]. In addition, most previous intervention studies examine medication education and medication compliance, and while some studies examined psychological counseling intervention for improving patients’ adherence to tuberculosis medication [7,10]. Hence, knowledge and information about care interventions that help resolve physical and psychological problems affecting patients with TB are still lacking.

Laughter is a behavioral response to stimuli such as jokes, positive emotions, pleasant thoughts, or cognitive perceptions of disharmony (dry chuckle) and a unique emotional response to joy [11,12,13]. Laughter therapy as a type of cognitive-behavioral therapy improves physical, mental, and social health and ultimately enhances QOL [11,12,13,14,15]. Currently, laughter therapies are divided into spontaneous and simulated laughter therapies [11]. Spontaneous laughter refers to intrinsic laughter triggered through stimuli such as humor and causes the contraction of ocular muscles [11]. Simulated laughter is defined as consciously induced laughter in a controlled environment without a specific reason instead of being caused by humor or other stimuli. Thus, simulated laughter is considered self-induced, fake, or humor-unaccompanied laughter. According to the theory that motion generates emotion, the human mind can differentiate between spontaneous and simulated laughter but the body cannot. Thus, laughter has therapeutic effects regardless of voluntariness, and the effects are similar to spontaneous laughter [11,16]. Simulated laughter therapy comprises various skills and exercises (e.g., dancing, singing, meditation, and yoga) at all stages [11,16]. Thus, the intensity and duration of simulated laughter therapy can be increased, producing additional health benefits [16], and the therapy is considered applicable in diverse populations and places [11]. A recent systematic review and meta-analysis confirmed that simulated laughter may improve depression, stress, and anxiety [16]. However, most studies on the simulated laughter effects had a high risk of bias due to an absent control group and small sample size, limiting the studies in confirming the effects of simulated laughter therapy. Furthermore, effects documented in previous studies were primarily focused on the psychological aspects, such as depression, stress, and anxiety, and understanding and evidence for the physical effects of simulated laughter remain lacking.

Thus, this study aimed to investigate the effects of simulated laughter therapy comprising various activities (breathing exercises, singing and dancing, and meditation) in TB inpatients facing possible psychological and physical health concerns. The hypotheses of this study are as follows:

**Hypothesis** **1:**
*The experimental group that underwent simulated laughter therapy will significantly improve pulmonary function compared to the control group.*


**Hypothesis** **2:**
*The experimental group that underwent simulated laughter therapy will have reduced physical symptoms compared to the control group.*


**Hypothesis** **3:**
*The experimental group that underwent simulated laughter therapy will have reduced depression symptoms compared to the control group.*


**Hypothesis** **4:**
*The experimental group that underwent simulated laughter therapy will significantly improve HRQOL compared to the control group.*


The study findings provide evidence for developing effective care interventions to promote the health of patients with TB and chronic conditions with complex health problems.

## 2. Materials and Methods

### 2.1. Study Design

This quasi-experimental study uses a nonequivalent control group pretest-posttest design, aiming to investigate the effects of simulated laughter therapy on physical and psychological health in inpatients with TB.

### 2.2. Study Participants

Patients aged 19–69 years, diagnosed with TB, who at that time received hospital care in a TB treatment facility, had a negative smear for Mycobacterium tuberculosis, a forced expiratory volume in one second (FEV1) of ≥50%, and Center for Epidemiologic Studies-Depression Mood Scale-10 (CESD-10) depression score of four or higher were recruited. Eligible participants who provided written informed consent to participate in the study were enrolled. Patients diagnosed with TB with chronic obstructive pulmonary disease (COPD), inpatients in the nursing care integrated service ward, and those of a foreign nationality were excluded. To ensure homogeneity in the inpatient environment between the experimental and control groups, two similar national TB treatment facilities, with similarities in hospital size and operation system, inpatient ward environment, and healthcare staffing were chosen. Additionally, to control the program’s effects on the control group, patients hospitalized at the facility chosen for the simulated laughter intervention were assigned to the experimental group, and those at the other facility were assigned to the control group.

The sample size was determined following the criteria for sample-size determination for clinical trials using the G*Power Analysis 3.1 program. With a significance level of 0.05, power of 0.80, and effect size of 0.80, the minimum sample size per group required for independent *t*-tests was calculated to be 26. Considering a 15% withdrawal rate, we enrolled 30 participants for each group, for a total of 60 participants. The final number of participants included in the analysis was 26 in each group for a total of 52 participants. Of 30 initially enrolled participants in the experimental group, three participants who missed six or more sessions of the 16-session simulated laughter-therapy program and one discharged patient were excluded, and of 30 initially enrolled participants in the control group, one participant who voluntarily withdrew from the study and three discharged patients were excluded.

### 2.3. Simulated Laughter Therapy

Laughter therapy was administered by a laughter-therapy expert with a grade two clinical laughter therapist license and a TB nurse with 20 years of career at a TB treatment facility and respiratory training. The laughter therapy of this study was designed as a 60-min program administered twice a week for eight weeks based on the framework of laughter therapy and existing empirical studies. The contents and composition of the program were validated by two laughter therapists and one psychiatric nursing professor. Laughter therapy consisted of various laughing activities, breathing exercises, stretching, meditation, and singing and dancing. The introduction comprised ten minutes of stretching to foster intimacy and induce muscle relaxation. The main part of the program comprised 40 min of activities, including singing and dancing (10 min), laughter therapy (20 min), and respiratory training (10 min). Singing has been proven to change moods by providing emotional stimulation and helping patients with weakened pulmonary function adjust to physical activities and significantly improve pulmonary function and QOL [17]. The singing and dancing activities were to one required song (“Living in style”) and two songs of the participants’ choice. The program facilitator provided four songs at every session for participants to choose two songs from the list. To prepare the list of four songs, the facilitator surveyed participants’ favorite songs in advance, and only popular songs were chosen to ensure all participants could easily sing along. For the laughter-therapy component, we used 23 types of laughter identified by previous studies that effectively reduced depression and improved pulmonary functions and chose the topics weekly. Thus, the 20-min laughter therapy for each session included primary laughter (“laughing with an open chest”) with two to three other types of laughter appropriate for the weekly topic. Ten minutes of breathing exercise included repetition of “clapping the chest—taking a deep breath—breathing through pursed lips,” effective in improving lung functions. “Chest clapping” contributes to lung health by removing lung secretions and cleaning out the airway [18]. “Deep breath” facilitates ventilation and alveolar oxygen exchange through sufficient relaxation and contraction of the diaphragm and respiratory muscles [18]. “Pursed-lip breathing” reduces the respiratory rate and helps maintain an open airway for an extended duration, such that the amount of air entering and exiting the lungs increases to reduce the residual volume. The American Lung Association recommends using pursed-lip breathing to treat obstructive pulmonary diseases, such as asthma, pulmonary fibrosis, and chronic obstructive pulmonary disease (COPD) since it simultaneously improves lung dynamics and respiration. Breathing exercises were performed using the same protocol at every session, and the participants were encouraged to perform the breathing exercise in their daily lives. The cool-down consisted of ten minutes of mediative breathing, including closing one’s eyes, taking a deep breath, and slowly breathing out while listening to quiet music, to relax the mind and stabilize vital signs, and the participants could express themselves after the meditation (Table 1).

### 2.4. Measures

Physical health was assessed based on pulmonary function and physical symptoms, and psychological health was assessed based on depression and HRQOL. The study instruments were used after obtaining permission from the developers or researchers who adapted the corresponding instrument.

#### 2.4.1. Pulmonary Function

Pulmonary function was assessed based on FEV1 measured using the diagnostic pneumatometer COPD-6 (Vitalograph Ltd., Ennis, Ireland, 2016), and the measurement error was ±2%. The measurement was taken following the COPD practice guidelines (2012), wherein the patient bites on a disposable mouthpiece in a resting state and breathes in and out to the extent possible. Three measurements were taken, and the highest value was used, wherein a higher value indicated better pulmonary function.

#### 2.4.2. Physical Symptoms

Physical symptoms were assessed using the 16-item TB symptom scale developed and validated by Song [19]. This instrument comprises nine symptoms of TB (e.g., coughing up mucus, dyspnea, weakness or fatigue, and hemoptysis) and seven symptoms of TB drug side effects, such as skin problems (itching, rashes, bruising, or yellow skin), stomach problems (upset stomach, nausea, vomiting, diarrhea, or loss of appetite), and lack of feeling or tingling in the hands or feet. Participants were asked if they had recently experienced 16 items of symptoms of TB and the side effects of TB drugs. There were four response options for each item: 1 = never, 2 = sometimes, 3 = frequently, 4 = always. A higher score indicates more physical symptoms. The reliability (Cronbach’s α) of the instrument was 0.85 in the study by Song and 0.82 in this study.

#### 2.4.3. Depressive Symptoms

In this study, we used the CESD-10, a short version containing 10 of 20 items in the CESD originally developed by Radloff. The CESD-10 was developed by Andresen et al. in 1994 and had a high agreement with the CES-D (kappa = 0.97, *p* < 0.001). Each item is rated on a four-point Likert scale ranging from 0 “extremely rarely” to 3 “most of the time,” and a higher score indicates a greater level of depression. The reliability (Cronbach α) of the scale was 0.91 during development and 0.86 in this study.

#### 2.4.4. Health-Related Quality of Life

HRQOL was measured using the international version two of the Medical Outcomes Study 36-Item Short-Form Health Survey Instrument (SF-36) developed by Ware and Sherbourne. The SF-36 contains 36 items and broadly comprises a physical component summary (PCS) and mental component summary (MCS). The PCS contains four subscales, with 19 items for physical functioning (PE), 4 items for role physical (RP), 2 items for bodily pain (BP), and 5 items for general health (GH). Similarly, MCS contains four subscales, with four items for vitality (VT), two items for social functioning (SF), three items for role-emotion (RE), and five items for mental health (MH). Each of the eight subscales uses a three-point or five-point rating scale, and the scores for each item are weighted based on their responses and subsequently summed. The total score is then converted to a score from 0 to 100. The reliability of the eight subscales during development ranged from 0.78 to 0.93. In this study, the Cronbach’s α was 0.78 for GH, 0.92 for PF, 0.92 for RP, 0.77 for BP, 0.84 for VT, 0.73 for SF, 0.92 for RE, and 0.82 for MH. The reliability for PCS and MCS was 0.79 and 0.86, respectively. The Cronbach’s α of the overall HRQOL scale was 0.91 in this study.

### 2.5. Data Collection

This study was approved by the Institutional Review Board (IRB) at Mokpo National University (MNUIRB-20181221-SB-019-02). Concurrently, we explained the purpose and data-collection method to the director of the hospital in which the study is conducted and the head of the nursing unit and obtained approval from the hospital’s IRB (IRB-398837–2019-RO2) before proceeding with the study. A baseline survey was conducted on the experimental and control groups using a self-report questionnaire about general characteristics, disease-related characteristics, depression, physical symptoms, and physical and mental HRQOL. Pulmonary function was assessed based on FEV1 by asking the participants to measure their FEV1 using COPD-6 (Vitalograph Ltd., Ennis, Ireland, 2016). At the posttest, after completing the eight-week lung-strengthening laughter-therapy program, the same questionnaire used at the baseline was administered to the experimental and control groups. Data were collected from 16 July to 11 September 2019, by two nurses visiting the participants who were educated and trained for the measurement of FEV1 and the survey. After baseline and posttest surveys ended, the participants were given a gift as a token of appreciation for their participation and cooperation.

### 2.6. Data Analysis

Participants’ general characteristics were analyzed with descriptive statistics, including actual number, percentage, mean, and standard deviation. Homogeneity between the two groups was tested using the chi-square test, Fisher’s exact test, and independent *t*-test. The normality of the data was analyzed using the Shapiro–Wilk test. The changes in the dependent variables were analyzed using the Mann–Whitney U test for non-normally distributed variables (depression) and paired *t*-test for normally distributed variables (pulmonary function, physical symptoms, and physical and mental HRQOL). The differences between the two groups before and after the intervention were analyzed with ANCOVA for heterogeneous variables (depression, physical symptoms) and Wilcoxon’s signed-rank test for homogeneous variables (pulmonary function, physical HRQOL, and mental HRQOL).

## 3. Results

### 3.1. Characteristics of the Experimental and Control Groups and Baseline Homogeneity

There were no significant differences in age ($\chi^{2}\;$= 0.94, *p* = 0.354), sex ($\chi^{2}\;$= 1.08, *p* = 0.601), marital status ($\chi^{2}$ = 0.39, *p* = 0.755), educational level ($\chi^{2}\;$= 0.39, *p* = 0.866), economic status ($\chi^{2\;}$= 4.11, *p* = 0.245), smoking status ($\chi^{2\;}$= 4.13, *p* = 0.099), drinking status ($\chi^{2}\;$= 4.05, *p* = 0.118), type of TB medication ($\chi^{2\;}$= 4.06, *p* = 0.132), duration of TB ($\chi^{2}\;$= 1.80, *p* = 0.079), and exercise ($\chi^{2}\;$= 4.38, *p* = 0.075) between the two groups at the baseline. Thus, the two groups are confirmed to have been homogeneous in their general characteristics (Table 2).

Before administering the intervention, the differences in the physiological (pulmonary function and physical symptoms) and psychological health statuses (depression and HRQOL) between the experimental and control groups were analyzed using an independent *t*-test (Table 3). There were no significant differences in pulmonary function between the experimental (72.85 ± 13.71) and control groups (67.31 ± 14.96) (t = 0.329, *p* = 0.569) and in HRQOL. Thus, the two groups were homogeneous concerning pulmonary function and HRQOL. In contrast, depression and physical symptoms significantly differed between the groups; the experimental group had a significantly higher physical symptom score (13.81 ± 7.66) than the control group (8.23 ± 3.91) (t = 9.32, *p* = 0.004). Depression was not normally distributed; hence, homogeneity was tested using the Mann–Whitney U test. At the baseline, the depression score was 11.08 (±6.99) in the experimental group and 6.54 (±4.64) in the control group, showing a significant difference (Z = −2.45, *p* = 0.014).

### 3.2. Effects of the Laughter-Therapy Program

After the simulated laughter therapy, the experimental group had significant changes in pulmonary function, physical symptoms, depression, and HRQOL, while the control group did not have any significant changes (Table 4). First, concerning physical health, the experimental group had a statistically significant improvement in pulmonary function from 72.85 (±13.71) before the intervention to 77.31 (±12.46) after the intervention (t = −3.77, *p* < 0.001). In contrast, the control group did not show a significant change between baseline (67.31 ± 14.96) and posttest (70.12 ± 14.78) (t = −0.65, *p* = 0.522). The significance of the differences in the changes between the two groups was analyzed, and the results confirmed that the changes attained after the intervention significantly differed between the two groups (F = 13.36, *p* < 0.001). In the intervention group, the physical symptom scores significantly decreased from 13.81 (±7.66) at the baseline to 5.88 (±3.50) after the intervention (t = 7.30, *p* < 0.001). In contrast, the control group did not show a significant change between baseline (8.23 ± 3.91) and posttest (7.54 ± 4.51) (t = −0.65, *p* = 0.522). The significance of the differences in the changes between the two groups was analyzed by ANCOVA, and the results confirmed that the changes attained after the intervention significantly differed between the two groups (F = 9.51, *p* < 0.001). The experimental group that underwent simulated laughter therapy had significantly improved pulmonary function and physical symptoms than the control group, and thus hypotheses 1 and 2 of the study were accepted. Regarding mental health, the experimental group had a statistically significant reduced depression from 11.08 (±0.99) before the intervention to 4.35 (±4.59) after (t = 10.46, *p* < 0.001). In contrast, the control group did not show a significant change between baseline (6.54 ± 4.64) and posttest (4.58 ± 5.83), although the score slightly decreased (t = 1.30, *p* = 0.205). The significance of the differences in the changes between the two groups was analyzed by ANCOVA, and the results confirmed that the changes attained after the intervention significantly differed between the two groups (F = 42.44, *p* < 0.01). In the intervention group, HRQOL significantly increased from 43.66 (±21.63) at the baseline to 63.08 (±16.81) after the intervention (t = −9.31, *p* < 0.001). In contrast, the control group did not show a significant change between baseline 75.59 (±14.54) and posttest 75.14 (±15.32) (t = 0.11, *p* = 0.910). The changes in HRQOL score attained after the intervention significantly differed between the two groups (F = 7.39, *p* < 0.01). The study also verified hypotheses 3 and 4 regarding the effects of mental health assessed with depression and HRQOL.

## 4. Discussion

This study was a nonequivalent control group pretest–posttest quasi-experimental study aiming to investigate the effects of an eight-week simulated laughter-therapy course in inpatients with TB. The results showed that the simulated laughter therapy effectively improved the physical and mental health of inpatients with TB. The significant findings are discussed below.

Unlike the control group, which did not show significant changes in pulmonary function, a crucial physical health indicator of inpatients with TB, the experimental group that received laughter therapy showed a significant improvement of (4.46%) FEV1. This is consistent with the study results that administered a 12-week laughter therapy focused on singing and dancing, wherein the experimental group showed a 3.48% increased FEV1 [20]. In contrast, pulmonary function did not improve in a study that applied a five-week exercise and laughter-therapy course (three times per week) in patients with pneumoconiosis [21] and a study that administered a two-week respiratory rehabilitation course and laughter yoga in outpatients with COPD [22]. The inconsistent findings on the effects of pulmonary function in simulated laughter therapy may be due to the differences in patients’ pulmonary functions. Whereas the study population of studies wherein patients showed improved pulmonary functions, including those in our study, were patients with TB or patients that did not undergo pharmacological treatment for respiratory disease or received a medical diagnosis, those with no improvement in pulmonary functions were patients with pneumoconiosis or COPD. According to a recent trend analysis of respiratory rehabilitation intervention studies among patients with COPD [23], 76.47% of 26 studies reported no significant improvement in clinical pulmonary function tests, including FEV1. This suggests that simulated laughter therapy cannot significantly improve pulmonary function in patients with COPD and pneumoconiosis because most of these patients have sustained irreversible lung injuries [23]. In contrast, simulated laughter therapy, including various activities, may effectively improve pulmonary functions in patients with TB whose lung functions can be restored with pharmacological treatment [5] or older adults without particular lung disease. This might be partially explained by the mechanism wherein loud and delightful laughter brings about various effects on the heart, circulatory system, and respiratory system [24,25] and influences the breathing pattern by adjusting inspiration and expiration, with a pause in between, thereby increasing the vital capacity [26].

Before the laughter therapy, we observed the greater levels of physical symptoms and depression observed in the experimental group compared to the control group. This may be caused by the difference in medication. Participants under the first-line TB drug treatment were 61.5% in the experimental group compared to 35% in the control group. Thus, to test the effect of laughter therapy on physical symptoms and depression, ANCOVA was used to control for the pre-existing differences in the scores of depressive symptoms and physical symptoms between experimental and control groups. Our breathing-exercise-added laughter therapy effectively reduced physical symptoms in patients with TB. The physical symptom score of the initial experimental group, which was higher than that of the control group, reduced significantly from the control group after the intervention. In addition, the results of ANCOVA confirmed that the differences in reduced physical symptoms between the two groups were significant. Physical symptoms were assessed based on 16 items, including pulmonary function symptoms, pain, fatigue, adverse drug reaction, and sleep disorder, and the experimental group showed significant changes in 15 of these symptoms, excluding hemoptysis (not shown in the tables of the study). The lack of change in hemoptysis may be due to none of the participants having hemoptysis at the baseline of the study. The control group showed no changes in the 16 items concerning physical symptoms. Since both groups were undergoing pharmacological treatment for TB, improved physical symptoms in the experimental group and not in the control group may be an outcome of simulated laughter therapy. Thus, considering previous empirical study findings that simulated laughter therapy reduced various physical symptoms [27,28], our findings support the physiological mechanism through which laughter increases the heart rate, blood pressure, and respiratory rate early after onset; however, when the laughing stops, a relatively short relaxation phase follows, during which production of cortisol, the stress hormone, declines [29], resulting in reduced blood pressure and increased general circulation, digestion, and oxygen saturation, helping reduce various physical symptoms of TB [14,25,30].

Our laughter-therapy program effectively reduced depression in inpatients with TB. This supports previous findings that laughter contributes to positively transforming negative mental health states, such as anxiety, stress, poor QOL, and depression [30], by stimulating serotonin secretion in the gastrointestinal tract and reducing cortisol production [29,31]. Specifically, the effects of our laughter therapy on depression were greater than that of simple laughter therapy used in previous studies. We compared the depression scores across studies by converting the scores based on a total score of 100% due to the use of different instruments across studies. While simple laughter therapy decreased depression by about 4–10% in adults with chronic conditions [32,33], laughter therapy, including other activities such as singing and dancing, and breathing exercises used in the previous study [34] and our study, reduce depression by 20% and 21%, respectively. Thus, we infer that simulated laughter therapy consisting of various activities may have greater physical, mental, and social health benefits than simple laughter therapy and suggest further empirical studies to directly compare effect sizes between simulated laughter therapy and simple laughter therapy. Most cases of depression experienced by patients with an illness are from weakened functions or physical symptoms caused by the illness [6]. Therefore, additional breathing exercises and singing and dancing that can improve weakened pulmonary function in our therapy may lead to a greater reduction in depression. Meanwhile, some previous studies [21,34] that applied laughter therapy with other, added activities to older adult patients did not observe significantly reduced depression despite administering the interventions for a similar duration to our study. These studies administered additional hand-movement activities and laughter therapy to inpatients of a long-term-care hospital for older adults [34] and breathing-exercise-added laughter therapy to older adult patients receiving outpatient care for COPD [21]. One significant difference between these studies and ours is the age of the study population. A recent meta-analysis reported that depression in older adults is more effectively reduced after simple laughter therapy than after laughter therapy with additional activities [35]. Because aging deteriorates physical and cognitive functioning, having older adults perform two activities simultaneously may overwhelm them, adversely impacting their motor and cognitive functioning [36]. Hence, it is necessary to design functional-activity-added laughter-therapy interventions considering patients’ age and disease-related characteristics and the anticipated effects of the intervention.

Our simulated laughter therapy also effectively increased HRQOL in patients with TB. The breathing-exercise-added laughter therapy led to a more significant improvement of QOL than laughter therapy without other activities. As the instrument used to measure QOL differed across studies, we compared the QOL scores by converting all scores based on 100%. Laughter therapy improved QOL from 61% to 73.9% in breast cancer patients [32] and from 62% to 63.5% in older adults with osteoarthritis [33]. While the improvement of QOL ranged from 1.5% to 12.9% in previous studies [32,33], our experimental group that received laughter therapy involving singing and dancing and breathing exercises showed a significant 19.7% improvement in QOL. This difference supports the mechanism through which laughter therapy realigns various aspects of work and life, promotes a directionality in life by producing similar brain-wave patterns shown by people who reached a true meditative state [15], and suggests, at least partially, that adding breathing exercise and singing and dancing components to laughter therapy further enhances participants’ QOL. According to a recent HRQOL prediction model for inpatients with TB, pulmonary function, physical symptoms, and depression are predictors of HRQOL [6], and poor QOL among patients with TB can be attributed to diminished physical functioning and consequent restrictions in activities and loss of function, impairing self-esteem, causing depression, and negatively impacts life overall. Thus, a breathing-exercise-added laughter program may increase physical and mental QOL scores better than simple laughter-therapy programs by increasing pulmonary function, reducing physical symptoms, and markedly diminishing depression. Finally, we should consider the possibility that the anti-inflammatory effect of TB drug treatment can contribute to the effect of improving pulmonary function, physical symptoms, depression, and HRQOL of simulated laughter therapy. A recent study observed improvement in depression and QOL after six months of anti-TB medication therapy without additional psychiatric treatment [37]. In addition, the absolute change in mental health in the experimental group was similar to that in the control group in this study. This may imply that the standard care and improvement of TB may improve the mental health of TB patients. Thus, we suggest the importance of widening therapeutic interventions considering other aspects such as laughter therapy along with more directly disease-related aspects. Given depression and lung-function decline are very common in patients with TB, we also suggest the preventative use of simulated laughter therapy for their physical and mental health problems.

This study is novel as it is the first study to examine the effects of simulated laughter therapy in inpatients with TB and examines its effects on both physical and psychological health. Nevertheless, this study has the following limitations. We could only examine the effects of the breathing-exercise-added laughter therapy on pulmonary function, physical symptoms, depression, and QOL and not the retention of the effects of the intervention. Further, while the baseline and posttest surveys were conducted on the same day for both groups, the specific time of the surveys differed. Therefore, the findings should be interpreted considering the variances in the time of measurement of pulmonary function, physical symptoms, depression, and HRQOL between the two groups. Finally, the study population was set to the inpatients of two national TB treatment facilities and almost 90% of participants were men; hence, the findings cannot be generalized to women and the entire TB population. Replication studies on patients with TB in other types of healthcare institutions and community-dwelling patients and further research on gender perspectives are needed. Unlike our study participants hospitalized in TB treatment facilities, TB patients hospitalized with other patients with non-infectious diseases or living in the community may suffer more from TB stigma. Therefore, it is expected that the mental health effects of laughter therapy for them will be different from the results of our study.

## 5. Conclusions

The laughter therapy administered in this study effectively improved pulmonary functions, reduced physical symptoms, diminished depression, and enhanced HRQOL in inpatients with TB. Thus, this study presents evidence supporting simulated laughter therapy’s use as a care intervention to alleviate physical and mental health problems at a TB-treatment facility. Administering the breathing-exercise-added simulated laughter therapy used in this study to inpatients with TB experiencing various physical and mental health problems will contribute to boosting the patients’ QOL and improving the TB cure rate.

## Figures and Tables

**Table 1 ijerph-19-10191-t001:** Simulated laughter therapy for hospitalized patients with pulmonary tuberculosis.

Phase	Activities
Warm-up (10 min)	▪ Ice breaking and stretching – Introduction and greetings – Face muscle and commissure of mouth stretching – Whole-body stretching
Main activities (40 min)	▪ Sing and dance (10 min) – Chorusing and movement, clapping or dancing with 3 songs per session – Basic song “Pom-na-gae Sal-ger-ya” – Choose 2 songs out of 4 preferred songs
	▪ Practicing various forms of simulated laughter (20 min) – Performing basic laughter with an open chest, breathing, and stretching – Choose 2~3 out of 23 laughter types per session
	▪ Performing laughter with the breathing exercise (10 min) – chest stroke: 10 times on each side – deep breathing 10 times – purse lip breathing 20 times
Cool-down (10 min)	▪ Meditation with calming music – Try to be calm and feel comfortable with soft music – Pause in every breathing cycle and be aware of it – Sharing feelings and saying goodbye.

**Table 2 ijerph-19-10191-t002:** Homogeneity of socioeconomic and health-related characteristics between experimental (*n* = 26) and control (*n* = 26) groups.

	Experimental Group *n* (%) or Mean (±SD)	Control Group *n* (%) or Mean (±SD)	*X^2^* or *t (p)*
Age (year)	55.92 (±10.83)	53.19 (±10.20)	0.94 (0.354)
Gender			1.08 (0.601) ^a^
Male	23 (88.5)	25 (96.2)	
Female	3 (11.5)	1 (3.8)	
Marital status			0.39 (0.755)
Single	6 (23.1)	8 (30.8)	
Married/others	20 (76.9)	18 (69.2)	
Education			0.39 (0.866) ^a^
Elementary school or less	6 (23.1)	5 (19.2)	
Middle school	6 (23.1)	5 (19.2)	
High school or more	14 (53.8)	16 (61.5)	
Subjective economic status			4.11 (0.245) ^a^
High	-	-	
Middle-high	3 (11.5)	3 (11.5)	
Middle	3 (11.5)	9 (34.6)	
Middle-low	5 (19.2)	3 (11.5)	
Low	15 (57.7)	11 (42.3)	
Smoking			4.13 (0.099) ^a^
Current/past smoker	20 (76.9)	25 (96.2)	
Non smoker	6 (23.1)	1 (3.8)	
Drinking			4.05 (0.118) ^a^
Current drinking	2 (7.7)	0 (0.0)	
Past drinking	17 (65.4)	23 (88.5)	
Non drinking	7 (26.9)	3 (11.5)	
Exercise			4.38 (0.075)
Yes	14 (53.8)	21 (80.8)	
No	12 (46.2)	5 (19.2)	
Type of medication			4.06 (0.132)
First-line drugs	16 (61.5)	10 (38.5)	
Second-line drugs	10 (38.5)	16 (61.5)	
Length of morbidity (months)	9.65 (±8.72)	6.23 (±4.31)	1.80 (0.079)

^a^ Fisher’s exact test; SD = Standard Deviation.

**Table 3 ijerph-19-10191-t003:** Homogeneity of pulmonary function, physical symptoms, depressive symptoms, and health-related quality of life between experimental (*n* = 26) and control (*n* = 26) groups.

	Experimental Group Mean (±SD)	Control Group Mean (±SD)	*Z**(p)* or *t (p)*
*Physical health*			
Pulmonary function (FEV1)	72.85 (±13.71)	67.31 (±14.96)	0.33 (0.569) ^a^
Physical symptoms	13.81 (±7.66)	8.23 (±3.91)	9.32 (0.004) ^a^
*Psychological health*			
Depressive symptoms	11.08 (±6.99)	6.54 (±4.64)	−2.45 (0.014) ^b^
Health-related quality of life	43.67 (±21.63)	75.59 (±14.54)	3.63 (0.062) ^a^

^a^ Independent *t*-test; ^b^ Mann–Whitney U test; SD = Standard Deviation.

**Table 4 ijerph-19-10191-t004:** Comparison of physical and psychological health between pre- and posttest among experimental (*n* = 26) and control (*n* = 26) groups and group differences in changes.

Variables	Group	Pre-Test Mean (±SD)	Posttest Mean (±SD)	*t* (*p*) ^a^	*F* (*p*) ^b^
*Physical health*					
Pulmonary function	Experimental	72.85 (±13.71)	77.31 (±12.46)	−3.77 (<0.001)	13.36 (<0.001)
	Control	67.31 (±14.96)	70.12 (±14.78)	−0.65 (0.522)	
Physical symptoms	Experimental	13.81 (±7.66)	5.88 (±3.50)	7.30 (<0.001)	9.51 (0.003) ^c^
	Control	8.23 (±3.91)	7.54 (±4.51)	−0.65 (0.522)	
*Psychological health*					
Depressive symptoms	Experimental	11.08 (±0.99)	4.35 (±4.59)	10.46 (<0.001)	42.44 (<0.001) ^c^
	Control	6.54 (±4.64)	4.58 (±5.83)	1.30 (0.205)	
Health-related quality of life	Experimental	43.66 (±21.63)	63.06 (±16.81)	−9.31 (<0.001)	7.39 (0.009)
	Control	75.59 (±14.54)	75.14 (±15.35)	0.11 (0.910)	

^a^ Paired *t*-test for mean differences between pre- and posttest of each group; ^b^ Group differences for changes between pre-and posttest; ^c^ ANCOVA statistics were calculated to control for the pre-existing differences on the scores of depressive symptoms and physical symptoms between experimental and control groups; SD = Standard Deviation.

## Data Availability

Not applicable.

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
