# Peer review of "Effects of Simulated Laughter Therapy Using a Breathing Exercise: A Study on Hospitalized Pulmonary Tuberculosis Patients"

_ijerph, 2022, doi:10.3390/ijerph191610191_

Round 1

Reviewer 1 Report

Overall impression

The topic is new and the manuscript covers it thoroughly so that this paper deserves publication. The research question is well defined, hypotheses are clearly stated, and the aims are well postulated. The study is of adequate size to investigate the research question, although I feel that the authors should have included an equal number of men and women (almost 90% of participants are women). The results suggest that when applying this kind of therapy, the psychophysical health of people suffering from tuberculosis improves

In the Data collection subsection, the authors wrote that they gave to the participants „a gift as a token of appreciation for their participation and cooperation“ two times, „after baseline and post-test surveys“. Please explain what kind of gift it was, is it typical of your culture?

Keywords – I suggest the authors to include the words „simulated laughter therapy“ and „tuberculosis“ here

Intoduction

Line 27 – please add an extra space between „Tuberculosis(TB)“

Lines 27-29 Please use consistent tenses of verbs in the same sentence; thus, change to „… 9.9 million people suffering from TB and 1.5 million dying from …“

Line 34 – there is an extra square bracket

Lines 43-44 Please add „… which is 1.5 times more than in the general population [7] and in patients …“

Line 72 – please change „is more effective on“ to „may improve“

Line 83 – please change „than the“ to „compared to the“

Please make the same change in the rows 85, 87 and 88

Materials & Methods

Line 98 – please change „currently receiving the inpatient care“ to „who at that time received hospital care in a TB treatment facility“

Line 99 – please change „are smear-negative“ to „had a negative smear“, „have forced“ to „a forced“

Results

Line 282 – an opening bracket is missing after the word „post-test“

Lines 285-287 Please rephrase this sentence, it is not clear what you meant.

Line 295 – there is an extra value here („5.88±3.50“); mean value, t and p-value here differ from the values given in Table 4

Lines 299-300 Please rephrase this sentence, it is not clear what you meant.

Discussion

In the limitations paragraph, please add that the majority of your sample were women.

Reviewer 2 Report

Searching for solutions to improve both physical and psychological symptoms and quality of life during treatment for tuberculosis is the main purpose of the authors that compare an intervention group with a control group. 

The document is well written, easy to read and clear.

There's an error in the word "meditation" in line 14

Introduction is informative regarding tuberculosis, particularly at Korea, and contextualizes

the use of laughter therapy, briefly reviewing the use and the possible mechanisms associated. 

Materials and Methods chapter is complete regarding study design, participants, intervention, statistical methodology. measures. The chapter is also informative regarding the Institutional Review Board approval.

Results are described both in text and using tables. 

Discussion is thorough and conclusions are aligned with the work results

As the intervention is more extensive than just the use of simulated laughter, eventually the title could be more aligned with the intervention content (laughter, entertaining, music related chorusing, breathing exercise and meditation), in the end it's difficult to isolate the individual impact of the different intervention components.

One doubt that I think it should be expressed is about the medication used, particularly the comparison between groups, 

In the discussion there's no mention of possible treatment impact on inflammation. From my point of view some consideration should be made to additional mechanisms that could explain the positive results. The same applies to depression symptoms and health-related quality of life. 

Besides being able to demonstrate the positive impact of the intervention the paper also makes a point regarding the importance of widening therapeutic interventions considering other aspects besides those more directly disease related.

It's worth reading 

Reviewer 3 Report

Tuberculosis(TB) is the leading infectious killer which threatens approximately ¼ of the world’s population. The Republic of Korea has the highest incidence rate of TB with the third-highest tuberculosis mortality rate among members of the Organisation for Economic Cooperation and Development (OECD). The present study sought to evaluate the health effects of simulated laughter therapy in hospital in-patients for TB. This quasi-experimental study included patients with TB from 2 different hospitals, with hospital A as laughter group and hospital B as a control group. The laughter therapy is an 8 week program held twice a week in a 60-minute group session consist of laughter, entertainment, music-related chorusing, breathing exercise, and mediation. The authors reported that laughter therapy decreased physical symptoms and increased pulmonary function. Moreover, the therapy also improved depression score and HRQOL. The authors concluded that simulated laughter therapy can help improving moderate depression, physical symptoms and aid the restoration of pulmonary function among patients with TB.

Specific comments:

11.       The exclusion criteria should be clearly stated in the methods.

22.       What could be reason(s) behind the greater levels of depression and physical symptoms observed in the experimental group, before the therapy? This should be included in the discussion.

33.       The absolute change (pre- vs post-) in the mental health in the experimental group looks very similar to that in the control group. Could this imply that mental health tends to improve with the progression of the trial, perhaps due to the standard care and/or improvement of the TB?

44.       There is also no description on the patient’s medication. Could the difference in medication contribute to the difference in levels of depression and physical symptoms before the therapy? This should also be included in the discussion.

55.       Given depression and lung function decline are very common in patients with TB, what about the preventative use of simulated laughter therapy on the related physical and mental health problems?

66.       “TB” was defined more than once and “tuberculosis” and “TB” were used interchangeably throughout the manuscript.

77.       OECD should be defined.

88.       Line 46, “…they have difficulty restoring the QOL of the general…” to “…they have difficulty restoring the QOL to that of the general…”.

99.       Please add “0” in front of the decimal point.

110.   Section 3.2, line 281-300, a number of the p values are different to table 4, eg. Line 280-281 “t=7.30, 280 p<.01”, table 4 “p<0.001”; Line 295 “…baseline to 5.88±3.50 after the intervention…”, table 4 says “63.06(±16.81)” after intervention; Line 295-296 “(t=-8.07, 295 p<.01)”, table 4 says “-9.31(<.001)”. Please double-check and verify.
